# Blood-Derived Lipid and Metabolite Biomarkers in Cardiovascular Research from Clinical Studies: A Recent Update

**DOI:** 10.3390/cells12242796

**Published:** 2023-12-08

**Authors:** Dipali Kale, Amol Fatangare, Prasad Phapale, Albert Sickmann

**Affiliations:** Leibniz-Institut für Analytische Wissenschaften-ISAS-e.V., 44139 Dortmund, Germany; amol.fatangare@isas.de (A.F.); prasad.phapale@isas.de (P.P.)

**Keywords:** cardiovascular disease, lipidomics, metabolomics, biomarkers, metabolipidomics

## Abstract

The primary prevention, early detection, and treatment of cardiovascular disease (CVD) have been long-standing scientific research goals worldwide. In the past decades, traditional blood lipid profiles have been routinely used in clinical practice to estimate the risk of CVDs such as atherosclerotic cardiovascular disease (ASCVD) and as treatment targets for the primary prevention of adverse cardiac events. These blood lipid panel tests often fail to fully predict all CVD risks and thus need to be improved. A comprehensive analysis of molecular species of lipids and metabolites (defined as lipidomics and metabolomics, respectively) can provide molecular insights into the pathophysiology of the disease and could serve as diagnostic and prognostic indicators of disease. Mass spectrometry (MS) and nuclear magnetic resonance (NMR)-based lipidomics and metabolomics analysis have been increasingly used to study the metabolic changes that occur during CVD pathogenesis. In this review, we provide an overview of various MS-based platforms and approaches that are commonly used in lipidomics and metabolomics workflows. This review summarizes the lipids and metabolites in human plasma/serum that have recently (from 2018 to December 2022) been identified as promising CVD biomarkers. In addition, this review describes the potential pathophysiological mechanisms associated with candidate CVD biomarkers. Future studies focused on these potential biomarkers and pathways will provide mechanistic clues of CVD pathogenesis and thus help with the risk assessment, diagnosis, and treatment of CVD.

## 1. Introduction

Cardiovascular disease (CVD) is a diverse group of heart and vascular disorders that affect a significant proportion of the global population. Despite recent significant advances in various drug- and device-based treatments, CVD is still recognized as one of the leading causes of morbidity and mortality worldwide, posing a substantial socioeconomic burden on individuals and populations [1]. According to statistics before the coronavirus disease 2019 (COVID-19) pandemic, CVD accounted for approximately 18.6 million deaths, representing about one-third of all deaths worldwide [1]. Moreover, the prevalence of CVD has increased by 17.1% from the previous decade, and it is projected to rise globally [1] continuously. The increase in CVD incidence is causally related to the increase in longevity, lifestyle, and environmental-related risk factors [2,3]. Thus, changes in particular lifestyle and environmental-related risk factors, and choices should improve the current situation. 

The increase in the prevalence of CVD may be linked to an increase in atherosclerotic cardiovascular disease (ASCVD) cases [4] since ASCVD, with its clinical manifestations including coronary artery disease (CAD), stroke, myocardial infarction (MI), and heart failure (HF), is the major contributing factor to all CVD fatalities [1,5,6,7]. It is a progressive and inflammatory disease characterized by arterial narrowing due to the gradual accumulation of lipids and endothelial dysfunction [8]. Dyslipidemia and hypertension [9,10] are the two main independent risk factors for ASCVD; other risk factors include insulin resistance, hyperglycemia, and obesity [11,12]. Numerous epidemiological and clinical studies [13,14,15] have established a relationship between aberrant blood lipid levels and the prevalence of CVDs such as CAD, MI, and stroke.

For decades, monitoring of the traditional lipid profile from blood tests has been used as a clinical diagnostic test for dyslipidemia, a prominent CVD risk factor which is characterized by elevated levels of low-density lipoprotein cholesterol (LDLC), triacylglycerols (TAG) [16,17], and low levels of high-density lipoprotein cholesterol (HDLC) [18,19]. Blood-derived biofluids, such as serum and plasma, are the preferred patient biospecimens for clinical cardiovascular research. Changes in traditional lipid profiles determine which patients with lipid disorders, and in the primary and secondary prevention of heart attack and stroke, would benefit from statin [20,21] or fibrate [22,23] therapy. Because CVD is a multifactorial disease, crude measurements of traditional lipids (TAG, LDLC, and HDLC) alone frequently fail to identify all individuals at risk for atherosclerotic plaque formation and subsequent thrombus formation [24,25]. This highlights the urgent need for the discovery of clinically applicable biomarkers for the prognosis, diagnosis, treatment, and monitoring of CVD.

Metabolites (including lipids) can be responsive to both genetic background and environmental exposure, thus directly providing the specific molecular phenotypes of the biological systems [26]. Their profiling reflects the pathophysiological processes underlying human diseases [27]. During the past decades, with the advances in analytical technologies, the new era of high-throughput disciplines such as metabolomics (small molecules with a molecular mass ≤ 1500 Da) for profiling of (hydrophilic) metabolites and lipidomics (usually considered a subset of metabolomics) for the profiling of (hydrophobic metabolites) lipids in a biological system have emerged.

This review provides an overview of MS-based lipidomics and metabolomics (metabolipidomics) workflows, with a focus on commonly used analytical techniques and strategies. Clinical metabolipidomics study typically includes a collection of clinical samples, lipid/metabolite sample preparation, MS data acquisition, statistical analysis, biological interpretation, cross-validation of biomarker candidates, and, finally, clinical translation of biomarkers, as illustrated in Figure 1.

This review summarizes promising CVD biomarkers in human blood plasma/serum samples from recent lipidomics and metabolomics studies. Then, potential CVD pathophysiological mechanisms associated with these biomarkers are described. The association of these potential biomarkers and pathways with CVD should be evaluated in future investigations.

## 2. Lipidomics and Metabolomics: Analytical Workflows and Techniques 

Among the many (>200) risk factors identified, an abnormal lipid panel (dyslipidemia) is the major morbidity associated with CVD. Lipids, which are essential components of cells, play three primary roles: signaling, energy storage, and structural support [29]. Plasma lipidome includes sterol lipids (cholesterol and CE), glycerophospholipids (PC, PC-O, LPC, PE, and PI), glycerolipids (MAG, DAG, and TAG), and sphingolipids (SM, DhCer, Cer, LacCer, HexCer, and FFA) as the most abundant lipids. They are frequently analyzed in blood-derived biofluids, and all cumulatively account for approximately 88% of the plasma lipidome [30] and more than 70% of the entire plasma metabolome [31]. The remaining plasma metabolome consists of hydrophilic metabolites such as carbohydrates, amino acids (involved in protein metabolism), nucleic acids (involved in DNA/RNA metabolism), organic acids, and acylcarnitines.

Lipid and metabolite analysis from complex biological samples such as plasma/serum requires dedicated sample preparation procedures and chromatographic separation modes due to the different physiochemical properties of these small biomolecules. A typical lipidomics workflow comprises sample preparation using the Bligh–Dyer [32], Matyash [33], or Folch [34] lipid extraction methods, whereas a metabolomics workflow involves sample preparation using polar organic solvents such as methanol, acetonitrile, ethanol, and their mixtures. These traditional sample preparation workflows for lipidomics and metabolomics require different aliquots of the same sample. In recent years, novel, simple, and time-saving sample preparation methods to simultaneously extract lipids and metabolites from the same sample, based either on biphasic [28,35] or monophasic [36] solvent systems, have been developed. These sample preparation workflows enable the parallel analysis of both lipids and metabolites from the same sample, thereby increasing the comparability of multiple “omics” datasets and limiting technical variability.

Most metabolipidomics studies rely on powerful analytical techniques such as NMR and MS for metabolite profiling and structural elucidation of unknown metabolites. Although NMR, being a non-destructive, unbiased, and robust technique, offers several advantages, it suffers from low sensitivity and can only detect up to 200 metabolites existing at higher concentrations (≥1 μM) in biological samples [37]. Compared to NMR, MS provides excellent sensitivity and detection limits (from the femto- to the atto-molar range) and a wide dynamic range of coverage (∼10^3^–10^4^). MS separates and identifies ions based on their mass-to-charge (*m*/*z*) ratio and, due to its fast scanning speed, it can simultaneously measure hundreds to thousands of analytes with high sensitivity in a single run [37].

In MS-based analyses, matrix-assisted laser desorption/ionization (MALDI) is a “soft” ionization technique that uses a laser energy-absorbing matrix to ionize analytes with minimal fragmentation. MALDI-MS imaging (MALDI-MSI) has been extensively used to visualize the in situ spatial distribution of metabolites from tissue samples [38,39]. Few MALDI-MS-based studies have investigated the serum metabolome profile associated with the clinical phenotype of the diseases [40,41]. The use of MALDI-MS for large-scale biofluid analysis is limited by the coverage of metabolite classes, background noise in the matrix signal, fluctuations in the analyte signal due to matrix “sweet spots”, and poor analyte quantification [40,42].

Electrospray ionization (ESI) is another ionization technique in MS that produces gas-phase ions from analytes in a solution using an electrospray. The ESI-MS interface has been extensively used in clinical research laboratories for routine analysis over several ionization methods such as electron impact (EI), chemical ionization (CI), or MALDI, etc. Biomolecules can be characterized using the (ESI-)MS technique, with prior chromatographic separation, or without prior chromatographic separation, i.e., by direct MS analysis of samples. Direct MS analysis relies on platforms such as native ESI/nano-electrospray ionization (nESI-)MS and flow injection analysis (FIA-)MS to increase samples throughput.

Hyphenated techniques such as liquid chromatography (LC-)MS (with an ESI interface) and gas chromatography (GC-)MS boast several advantages, such as the minimization of ion suppression/enhancement effects, reduction in spectral overlap, and, thus, decreased spectral complexity. Gas chromatography (GC), with an electron ionization (EI) or chemical ionization (CI) interface, is often used for the separation of volatile and thermally stable compounds. GC-MS offers, for low-abundant key metabolites such as BCAA, polyamines, glycolysis, and TCA pathway intermediates (such as pyruvate, fumarate, and citrate) [43,44,45], superior peak resolution and higher sensitivity, in comparison to LC, often employing derivatization to generate more volatile analytes. However, GC-MS is not suitable for separating species with insufficient volatility or thermal instability, such as acylcarnitines (AC) or acyl-coenzyme A (CoA) found in biofluids [46]. 

LC can separate a wide range of chemically diverse sets of compounds, which makes it the method of choice for routine analysis in clinical research laboratories. Correspondingly, many recent studies implement LC-MS for the comprehensive analysis of polar metabolites and non-polar lipids in CVD research [46]. In general, polar metabolites are separated on hydrophilic interaction liquid chromatography (HILIC), and non-polar lipids are separated on reverse phase (RP-)LC [47,48]. A strategy for dual separation of both polar metabolites and non-polar lipids using a two-dimensional LC(-MS) system combining HILIC separation followed by RPLC separation allows for the separation of both lipids and metabolites from the same extract in a single injection [49,50,51]. In particular, these technological advances support large-scale clinical trials with limited sample volumes and reduce the required device run time.

In general, MS-based metabolipidomics analysis can be performed using targeted or untargeted approaches (Figure 2), with each approach having its own advantages and limitations. While the former is a hypothesis-testing approach, the latter is a hypothesis-generating approach involving the discovery and validation of novel candidate biomarkers; however, the process is labor-intensive. The targeted analysis mainly focuses on analyzing a limited set of pre-selected target compounds (tens to hundreds) based on extensive biological hypotheses [52]. It can be performed on both high-resolution (mass resolution ≥30,000 at 200 Da) MS (HRMS) instruments and low-resolution MS (LRMS) instruments. The targeted quantification of compounds by selected/multiple reaction monitoring (SRM/MRM)- or parallel reaction monitoring (PRM)-based data acquisitions enables the accurate and reproducible quantification of analytes.

Untargeted analyses simultaneously measure a broader range of detectable (thousands) features and cover the detection of “unknown metabolites” in the biological sample. For this purpose, HRMS instruments based on mass analyzers such as the Fourier Transform Ion Cyclotron Resonance (FTICR), quadrupole orbitrap (Q-orbitrap), and quadrupole time-of-flight (QqTOF) are utilized. Often in these analyses, MS instruments first perform a full scan MS (MS1), and subsequent tandem MS (MS/MS or MS2) scans are obtained in the data-dependent acquisition (DDA) or data-independent acquisition (DIA) modes. All features obtained are processed and statistically analyzed so that differentially expressed features can be later identified. Feature identification often relies on matching MS1 and MS2 spectra and retention times to spectral libraries or by running reference standards in the same chromatographic run. This approach is often limited by the molecular diversity and complexity of the features detected, complicating subsequent data analysis and interpretation. 

Non-targeted assays are commonly used for the high-throughput screening of small molecules in complex biological specimens. Despite their proven utility in the measurement of diverse molecules simultaneously, these assays are known to overlook low-abundant metabolites [53] (e.g., signaling lipids such as phosphatidyl inositol phosphates (PIPs), and eicosanoids and its metabolites). The analysis of such low-abundant metabolites needs targeted analysis, and may sometimes require additional selective sample enrichment/purification steps during the sample preparation procedures prior to MS analysis. In recent years, for the analysis of low-abundant metabolites, several new MS-based analytical methods and techniques (e.g., [54,55]) have been developed. In the future, such analytical methods could be used to comprehensively characterize the homeostasis of these low-abundance analytes under health and disease conditions.

A hybrid approach that integrates both untargeted and targeted approaches will provide comprehensive signatures of the human CVD metabolome (Figure 2). Although MS is a powerful tool for the structural elucidation of complex biological samples, it has a few limitations. For example, the distinction between multiple isomeric, or enantiomeric compounds [56] with subtle structural differences in the same coelution profile is not possible. With emerging technologies in MS such as ion mobility-mass spectrometry (IM-MS), it is feasible to separate isomers rapidly [57]. However, many advancements are to be seen in the mainstream applications of these technologies.

In recent years, several lipidomics and metabolomics studies involving large-scale epidemiological and clinical cohorts based on different analytical and methodological approaches have proposed numerous biomarkers for CVD. There is an urgent need to leverage the knowledge from current metabolipidomics studies to identify promising biomarker candidates and translate them into clinical applications. Accordingly, we aimed to summarize the CVD biomarkers (Table 1) from studies published over the last 5 years in this review. Several keywords such as “biomarker”, “metabolomics”, “lipidomics”, and “cardiovascular disease” were searched in PubMed. Reviews and animal-based research articles were excluded, and only studies based on human plasma/serum using MS analysis were included. 

## 3. Lipid and Metabolite Biomarkers of CVD

MS-based lipidomics and metabolomics studies have shown the potential to screen biomarkers for human diseases [83,84,85]. To date, ceramides or ceramide panel combinations are the only biomarkers that are translated from metabolipidomics research into clinically approved laboratory tests. These bioactive lipids are central components of sphingolipid cell signaling pathways. They play myriad roles in cellular processes such as the regulation of apoptosis [86], cellular stress, and vascular inflammation [87,88,89,90]. Cardiometabolic diseases such as CAD [91,92] ischemic stroke [93], HF [94], lipotoxic cardiomyopathy [95], and MI [96] are associated with elevated circulating plasma ceramide levels. Higher ceramide levels are also implicated in major comorbidities of CVD, namely, type 2 diabetes mellitus (T2D) [97,98] and insulin resistance [99].

Ceramides are mostly generated in the liver, and transported into the bloodstream by lipoproteins. Two primary pathways produce ceramide (or dihydroceramide, in the event of de novo synthesis): a de novo synthesis (beginning with fatty acids) and re-acylation of sphingosine (salvage pathway), both regulated by ceramide synthases (CerS). Mammals have six isoforms of CerS (CerS1-6), each of which can preferably synthesize ceramides with a specific fatty acyl chain length (from 14 to 26 carbon atoms) [91,92], and expression patterns of CerS vary across different tissue types. Numerous studies have implicated that each CerS species has specific pathophysiological roles [100,101,102,103].

Several studies have been carried out to verify the functions of specific isoforms of CerS, and CerS6 expression in particular has been correlated with insulin resistance. Turpin et al. [101] have shown that mice lacking CerS6 were protected from glucose intolerance and high fat, diet-induced obesity. Increasing evidence suggests that CerSs are potential therapeutic targets to treat CVD, by modulating specific ceramide levels in various pathologies.

The correlation between alterations in biomarkers in the biofluids and the cellular/tissue environment of patients with CVD may provide significant insights into the role of metabolic dysfunction in the progression of CVD pathophysiology. This correlation between biofluids ceramide metabolite markers and cellular/tissue alterations offers promising avenues for identifying potential diagnostic biomarkers, therapeutic targets, and personalized treatment strategies, thereby advancing precision medicine approaches in managing and combating cardiovascular diseases. 

Given the critical role of ceramides in the development of CVD, several analytical methods have been developed and validated to quantify ceramides in human plasma and serum [104,105,106] samples.

Various research studies mentioned later have shown that clinical plasma/serum measurements of ceramides provide an independent added value to the routinely used diagnostic and prognostic tools for CVD events. The first clinical study to introduce the association of plasma ceramide levels with CVD and total mortality in CAD disease is the LURIC (Ludwigshafen Risk and cardiovascular health) [107]. Thereafter, three independent CAD cohorts, namely the Special Program University Medicine Acute Coronary Syndromes and Inflammation (SPUM-ACS) and the Bergen Coronary Angiography Cohort (BECAC) studies [108], have reconfirmed and validated the use of ceramides as predictors of CV mortality. In other studies, high levels of plasma/serum C16:0, C18:0, and C24:1 ceramides, and ratios of these ceramides to C24:0 ceramide, were shown to predict adverse cardiac events in healthy individuals and patients [108,109,110]. Further, ZoraBiosciences developed the Cardiovascular Event Risk Test (CERT)-1 risk score, comprising previously stated plasma/serum ceramide or ceramide ratios, which was used to stratify individuals into four (low, medium, increased, and high) risk categories for the primary and secondary prevention of CVD. Following this, the first diagnostic test to assess the risk of cardiovascular events in CAD patients was launched by the Mayo Clinic in the USA. In the Western Norway Coronary Angiography Cohort (WECAC) study, the CERT-1 risk score prediction was improved by including distinct phosphatidylcholines (PCs), termed the CERT-2 risk score. The CERT-2 score was verified in the LIPID and KAROLA studies [58]. Further, the sphingolipid-inclusive CAD risk score, termed the SIC score [59], for demarcating CAD patients compared to controls, and the diabetes score (dscore) [111] for predicting the onset of diabetes, have been developed. 

TMAO (trimethylamine N-oxide), a circulating gut microbiota-dependent metabolite mainly derived from the oxidation of trimethylamine (TMA) and its dietary precursor [112] choline (derived from dietary PC or the hydrolysis of endogenous PC), has recently emerged as a potential biomarker for CVD. Since its discovery [112], TMAO has been implicated as a predictive biomarker for thrombosis and platelet hyperactivation [113], increased CVD risk, and atherosclerotic plaque formation [114,115,116,117] in many clinical studies. Elevated levels of TMAO in plasma are independent biomarkers of CVD for secondary prevention. The Cleveland Clinic Heart Lab commenced blood tests to measure blood TMAO levels using LC-MS/MS [118]. Although there is a broad consensus that elevated plasma TMAO levels are independent CVD biomarkers for secondary prevention [119], few recent studies [120,121] provide conflicting evidence for an association between circulatory TMAO metabolism and CVD events, suggesting that TMAO is not an independent risk factor for evaluating CV risk events. Thus, currently, traditional lipid panels, ceramide panels, and TMAO are the only established screening tests to assess future risks of CVD. Elevated levels of acylcarnitines (the transport form of activated FAs) and BCAAs are associated with an increased risk of CVD events [46,122,123,124]. This dysregulation causes a mitochondrial imbalance between mitochondrial fatty acid β-oxidation (MFAO) and glucose oxidation. Fatty acids serve as an energy source in MFAO, and acylcarnitines, which act as long-chain fatty acid carriers into the mitochondria, play important roles in mitochondrial energy homeostasis through MFAO [125,126]. MFAO is affected by BCAA catabolism in which the excess catabolic flux of BCAAs causes the accumulation of trans-endothelial fatty acid into the muscle, the accumulation of lipotoxic, incompletely esterified intermediates such as DAG, and blunted insulin signaling [127,128], thus effectively clogging the β-oxidative machinery in a manner analogous to the effect of excess fat [129]. Furthermore, both fatty acid oxidation and BCAA catabolism are closely interconnected through energy metabolism, as both can affect glycolysis through inhibiting pyruvate dehydrogenase and the TCA cycle through acetyl-CoA/succinyl-CoA/substrate (“cycle”) overload, as well as mitochondrial oxidative phosphorylation through their common target, PGC-1α [126]. These overall changes in various lipids, fatty acids, and BCAAs in circulation emphasize the disturbances in metabolic homeostasis and energy metabolism in CVD [126,130,131]. Furthermore, clogging of β-oxidation machinery may explain the decreased glucose utilization, increased glucose tolerance, and the development of insulin resistance in muscle cells, relevant for T2D [129] pathophysiology, which is a comorbidity often associated with CVD. This also highlights the role of the accumulation of BCAA in failing hearts, which was associated with myocardial insulin resistance [44,46]. Altogether, abnormal levels of circulating FAs, acylcarnitines, and BCAAs modulate insulin sensitivity, energy metabolism, and inflammation, and thus contribute to CVD risk factors such as T2D and obesity.

Tryptophan degradation via the kynurenine pathway is an important pathway in energy metabolism, leading to the biosynthesis of nicotinamide adenine dinucleotide (NAD+) [132]. Kynurenine pathway intermediates and enzymes have been proposed as inflammatory markers [133,134] in CVD [135,136]. The kynurenine pathway and CVD risk factors such as hypertension, diabetes mellitus, dyslipidemia, and obesity seem to be at play here [133,137]. Accordingly, several studies already note kynurenine pathway metabolites as diagnostic or predictive markers of ASCVD or CAD [reviewed in [137].

Along with the above pathways, inflammation, reactive oxidation species, and nitric oxide also play a role in CVD [138], however, it is difficult to capture these pathways’ alteration as the pathway intermediates are not easily detected in metabolomics analysis. In many reported studies, the mechanistic understanding of the role of newly screened biomarkers in the biological pathways underlying CVD has not yet been established or validated, and thus, they have limited clinical utility in CVD. After the discovery phase, these biomarkers should be validated and fulfill the proposed criteria for “biomarker qualification” before translating into clinical practice [139]. The addition of these biomarkers to the base model of CVD, along with other established biomarkers, could improve the mechanistic understanding of CVD as well as CVD-associated risk modeling.

## 4. Clinical Translation of Metabolite Biomarkers

Metabolites play diverse roles as cellular end products or byproducts, signaling molecules, enzyme modulators, and reactive species. Due to the advanced nature of these molecular indicators, mechanistic evaluation of their intended purpose with respect to clinical utility is complex, making clinical translation of metabolomic biomarkers a multifaceted process.

Pre-analytical biases in clinical “omics” can significantly affect the overall reliability of study results and the suitability of specific lipids/metabolites as biomarkers. They arise primarily from differences in sample labeling, handling, collection, processing, or storage. For example, differences in sample treatment before the analysis of blood samples affected the quantification of blood-based metabolites [28,29,30]. Reliable quantification of lipids and metabolites in clinical studies can only be achieved by standardizing pre-analytical variables. Recent reviews [31,32,33,34] have proposed several recommendations for the standardization of preanalytical sample handling procedures to obtain reliable and reproducible results. The processing and analysis of obtained MS data and its biological interpretation, including integration with other omics/clinical data and pathway analysis and the clinical translation of biomarkers, have been extensively discussed in other reviews [35,36,37] and are not the focus of this review.

For assessing metabolomic biomarkers to validate their clinical utility, performance evaluations through comparative studies (including longitudinal and population-based studies) using established standards and biobank samples is necessary. Such investigations require improved accuracy, precision, and critical data analysis. Analytical quality criteria include precision, precision, sensitivity, specificity, wide dynamic range, and robustness in biomarker assays, as well as traceability to reference standards and robustness to sample variation. In clinical biomarker discovery research, validated analytical methods must be rigorously implemented to ensure that they are suitable for the intended purpose. Thus, in our opinion, the pertaining FDA ICH M10 guidelines should also be followed for biomarker discovery research. Additionally, regulatory eligibility for qualification for biomarker research will expedite the international harmonization of the biomarker search process.

In high-throughput untargeted metabolomics, metabolites are often putatively annotated by database searches based on the spectral matching of high-resolution MS1 and MS2 data. The putative annotations of metabolites can lead to the misidentification of biomarkers and, thus, to wrong biological interpretation [56]. The Metabolomics Standards Initiative (MSI) has proposed four confidence levels for annotating and identifying metabolites [140], with level one being the highest confidence level. However, only a few untargeted metabolomics studies have reported the identification (as proposed by MSI) of metabolites at level one [141]. Thus, the correct annotation of candidate biomarkers remains a major bottleneck in large-scale clinical metabolomics studies based on untargeted approaches.

The critical evaluation, integration, and analysis of MS-based metabolomics data is crucial for the successful clinical translation of metabolite biomarkers. The prognosis and diagnosis of cardiovascular disease (CVD) may now be predicted using machine learning, which presents promising possibilities for individualized treatment plans. These computational advances should be well utilized to facilitate the broad applicability of metabolomics datasets and multi-omics data integration.

The number of metabolomics studies has increased over the past decade, necessitating the transparent reporting of the results of these studies. This will help to increase comparability across different studies and could facilitate the harmonization of biomarkers (reviewed in [142,143,144,145,146,147]). Furthermore, efforts have to be made by research communities to make minimum reporting checklists mandatory components for both lipidomics [148,149] and metabolomics studies [150].

In conclusion, the clinical translation of metabolomic biomarkers in life science research demands a comprehensive understanding of their clinical utility and pathophysiological mechanisms. Investigating their performance metrics, evaluating their complementary value in refining diagnoses and prognoses, and assessing their direct influence on decision-making processes are pivotal steps toward unlocking the full potential of metabolites. Addressing these crucial questions is paramount for propelling the field towards an integrated application of metabolomic insights within clinical practice, thereby shaping the future landscape of diagnostics and personalized medicine.

## 5. Conclusions

Metabolipidomics studies on blood-based biofluids have demonstrated the potential to pioneer the discovery of ceramides as clinical markers for the diagnosis and prognosis of CVD. Novel sample preparation strategies, automated data analysis workflows, machine learning algorithms, and faster MS instrumentation technologies have guided the rapid, robust, and large-scale quantification of lipids and metabolites. Comprehensive metabolomics and lipidomics analysis, in combination with proteomics and genomic analysis, can provide a complete “snapshot” of the pathophysiological mechanisms underlying CVD. This would consequently allow us to accurately identify clinically translated biomarkers and therapeutic targets of CVD. In this review (Table 1), we have noted the association of intricately interconnected biochemical processes such as fatty acid and carbohydrate, BCAA and gut microbial metabolism, glycerophospholipid, and sphingolipid pathways is reflected in many studies [58,59,60,63,65,68,70,76,78,79] where multiple metabolite classes are altered simultaneously in concerted fashion. The comprehensive monitoring of alterations in these pathway metabolites in future clinical studies as a whole (panel) will establish the causal or correlational association between CVD and metabolite biomarkers. 

## Figures and Tables

**Figure 1 cells-12-02796-f001:**
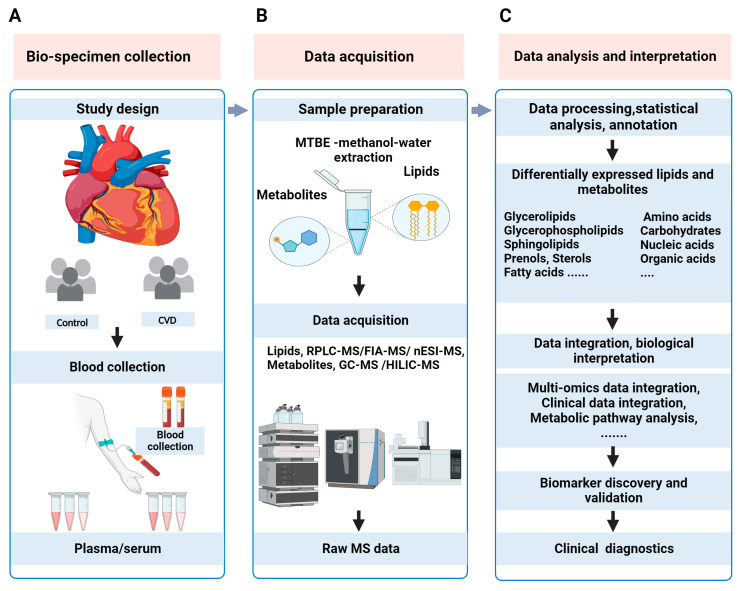
The typical workflow of MS-based clinical metabolipidomics analysis in CVD research. (**A**) Pre-analytical phase for clinical metabolipidomics study includes planning of study design, collection of blood samples, and separation of plasma/serum from whole blood. (**B**) Lipids and metabolites are extracted from plasma/serum samples, and then subjected to MS-based data acquisition. A schematic representation of the biphasic separation of lipids (top phase) and metabolites (lower phase) using the SIMPLEX [28] method is shown as a reference. (**C**) Data analysis is performed to obtain differentially expressed features. Further, outcomes of metabolipidomics data can be integrated with clinical or other “omics” data and interpreted biologically to provide insight into the pathophysiological mechanisms of CVD. After cross-validation, these biomarkers can be clinically translated for routine applications.

**Figure 2 cells-12-02796-f002:**
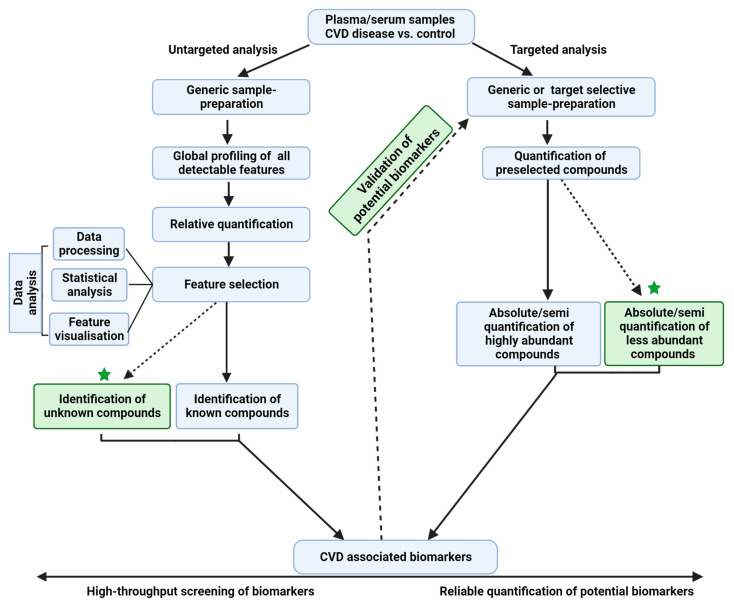
Overview of targeted and untargeted approaches used in metabolipidomics. The untargeted approach offers an unbiased approach for identifying novel biomarkers and generating new hypotheses related to CVD biology, while the targeted approach is useful for validation of these hypotheses and for quantification of low-abundant metabolites. A comprehensive analysis of the metabolome using a combination of these approaches will help to further our understanding of metabolite–CVD associations.

**Table 1 cells-12-02796-t001:** Summary of promising lipid and metabolite biomarkers recently discovered from human CVD studies using different analytical approaches.

Analytical Technique	Analytes of Interest	Potential Biomarker	Disease	Case, *n*	Controls, *n*	
**1.** **Targeted analysis approach**						
LC-MS	Ceramides, phospholipids	PC-16:0/22:5, 14:0/22:6, 16:0/16:0; Cer-C16:0, C18:0, C24:1, C24:0	CVD in ASCVD	340 ^#1^	3449 ^#1^	[58]
				503 ^#2^	5448 ^#2^	
				105 ^#3^	918 ^#3^	
	Sphingolipids (SPL)s	dhCer C18:0, Cer C22:0, sphingosine, dhSM 24:1, SM- C24:0, C18:0	CAD	462	212	[59]
	Abundant lipids	LPC 17:1, AC 18:2, TAG 51:0	STEMI	80	50	[60]
	AAs	4-Hydroxyproline	CAD	128	64	[61]
	Sphingosine 1-phosphate	Sphingosine 1-phosphate	HF	210	54	[62]
	TMAO, choline, creatinine, carnitine		CAD	302	59	[63]
	AAs, ACs, polar metabolites	AF: AC- C16:0, C18:0, C18:2	HF	326	426	[64]
		HF: AC- C14:0, C16:0, C18:0, C18:2	AF	509	617	
	ACs, BCAAs	AC-C3:0, C4:0, C4:0-OH, C5:0, C5:1-O2, C5:1, C5:0-OH, C6:0, C8:0, C8:0 DC, C8:1, C10:0, C10:1, C12:0, C12:1, C14:1, C16:1, C16:0, C18:1, C20:4; carnitine, valine, isoleucine	CAD	54	116	[65]
Direct MS	DhCer, Cer ^a^	DhCer C22:2, C26:1; Cer C16:0, C20:0	CVD	551	1137	[66]
	ACs	Acetylcarnitine (AC-C2:0)	CVD in T2D	356	676	[67]
	Abundant lipids ^a,b^	CE-20:(0 to 3), 22:(0 to 4), 24:0, 24:1; FFA- 15:0, 16:0, 18:4, 20:0, 20:3, 20:4, 22:1, 24:0, 24:1; MAG- 14:0, 15:0, 20:0, 18:1, 18:2, 18:3, 20:3, 20:4; 20:5; FAs: DAG 16:0, 18:0, 22:4; TAG 18:0; PE 16:1, 18:0, 22:4, P-22:4; P-22:5; Cer 14:0; dhCer 22:2; PC 20:2; PI 22:4; LacCer 16:0, 18:0, 18:1, 20:1, 26:0; HexCer 18:1, 24:1	CVD	551	1148	[68]
	ACs	AC- 14:0, 12:0, 14:0 DC, 14:0-OH, 20:0, 18:0, 16:1-OH, 8:0, 6:0, 10:0	DCM	50	50	[69]
	Abundant lipids ^c^	PC 15:0_18:2, 17:0_20:3, 16:0_20:1, O-16:2_18:0; SM 34:1; CE 18:0, DAG 18:1_18:3; PI 16:0_20:4	CAD	536	3329	[70]
LC-MS, Direct MS	AAs, biogenic amines, glycerophospholipids sugars, ACs, SPLs ^d^	PC O-40:6, O-38:6, 38:6, 38:5, 40:6	CAD	2166	8575	[71]
GC-MS	FAs	FA 20:0, 22:6n-3	ASCVD in T2D	44	74	[72]
**2.** **Untargeted analysis**						
LC-MS	Urobilin; SM 30:1		Incident HF	74 ^#4^ 183 ^#5^ 84 ^#6^	920 ^#4^ 1121 ^#5^ 1797 ^#6^	[73]
	9-Decenoylcarnitine		Prevalent AF	38 ^#4^ 20 ^#5^ 35 ^#6^	897 ^#4^ 1118 ^#5^ 1935 ^#6^	[74]
	Diosmetin, α-Ergocryptine, Fludrocortisone acetate, Quinic acid, 2,6-ditert-butylbenzoquinone		AF with HF	20	20	[75]
	5-oxo-D-proline, LPC 20:4, creatinine, LPE 16:0		ACS	284	130	[76]
	SM 18:0/14:0, LPC 18:0/0:0, 1-Methylpyrrolidinium, PC18:2/18:2, 18:0/18:0		IS	Discovery:		[77]
				120	40	
				Validation:		
				160	80	
	LPC 18:2, CE 18:1, alanine, choline, fructose		HF	87	49	[78]
LC-MS, GC.GC-MS	4-hydroxyphenyl acetic acid, Threonine, PC 36:2, PC-O 34:2, 34:3, SM 34:1		CVD in T1D	95	669	[79]
**3.** **Untargeted and targeted analysis**						
LC-MS	Indole-3-Propionic Acid		CAD ASCVD	30	30	[80]
				256	256	
	Trimethyl-5 amino valeric acid		Incident HF	1647		[81]
	Orotidine ^e^		CVD in T2D	Discovery:		[82]
				113	112	
				Validation:		
				300	299	

Abbreviations: n—number of participants. Profiling based on commercial metabolomics platforms: ^a^ Complex lipid panel™ Metabolon (Metabolon Inc., Morrisville, NC, USA); ^b^ Lipid classes containing specific FA chain length; ^c^ Lipotype shotgun lipidomics (Lipotype: Dresden, Germany); ^d^ Biocrates (MxP^®^ Quant 500 kit) (Biocrates, Innsbruck, Austria); and ^e^ Global discovery panel™ Metabolon (Metabolon Inc., USA). Clinical study names: #1: WECAC; #2: LIPID; #3: KAROLA; #4: PIVUS; #5: ULSAM; #6: TwinGene.

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
