# Peer review of "Blood-Derived Lipid and Metabolite Biomarkers in Cardiovascular Research from Clinical Studies: A Recent Update"

_cells, 2023, doi:10.3390/cells12242796_

Round 1
Reviewer 1 Report
Comments and Suggestions for Authors
Blood-derived lipid and metabolite biomarkers in cardiovascular research from clinical studies: a recent update
The title and abstract effectively capture the essence of the manuscript. The review is thoughtfully organized, covering various analytical techniques, workflow, and metabolites relevant to cardiovascular disease.
The paper is well-founded in a comprehensive literature review that encompasses studies in metabolomics and lipidomics, with an emphasis on methodology. This methodology is presented alongside its strengths and weaknesses.
Lines 232 and 241: correct the word metabolipiodmics
I suggest some changes that could make the manuscript easier to read.
In section 2, "Lipidomics and metabolomics: techniques and workflows," the authors first discuss the sample preparation workflow (which differs from the one depicted in Figure 1) and then the techniques most important in metabolipidomics. Perhaps this order of presentation could be reflected in the section's title.
In the same section, there is a mention of two types of metabolomics analysis: targeted and untargeted approaches. It would be beneficial to highlight that while the former involves hypothesis testing and finds application in clinical settings, the latter serves as a hypothesis generator and demands validation, making it a more labor-intensive process.
Between lines 210 and 215, the authors made a statement regarding the lack of established associations between low-abundant metabolites, such as DolPs and PIs, and diseases, but they did not provide any references to support this claim. Furthermore, it remains unclear why these metabolites were mentioned. If the intent was to demonstrate how new techniques and technologies can enhance the sensitivity of methods used to quantify low-abundant metabolites, the phrase needs to be redone.
Table 1 is currently challenging to read and should be redesigned. For instance:
|
Disease |
References |
Analytical Platform |
Biomarkers |
Findings |
|
CVD |
WECAC Study [106] Lipid Study Karola Sudy |
LC-MS |
Ceramide and PC |
Ceramide-PC score as predictor of residual events risk in patients with CAD |
Continuing with Table 1, diabetes is one of the listed conditions. Given that the title pertains to cardiovascular diseases, it would be better to eliminate these mentions from the table.
There is no connection between the content found on page 8, lines 308-314, and the content located on page 11, lines 326-339.
Author Response
The title and abstract effectively capture the essence of the manuscript. The review is thoughtfully organized, covering various analytical techniques, workflow, and metabolites relevant to cardiovascular disease.
The paper is well-founded in a comprehensive literature review that encompasses studies in metabolomics and lipidomics, with an emphasis on methodology. This methodology is presented alongside its strengths and weaknesses.
Lines 232 and 241: correct the word metabolipiodmics
→ Thank you for noticing this typing mistake. This has been corrected throughout the manuscript.
I suggest some changes that could make the manuscript easier to read.
In section 2, "Lipidomics and metabolomics: techniques and workflows," the authors first discuss the sample preparation workflow (which differs from the one depicted in Figure 1) and then the techniques most important in metabolipidomics. Perhaps this order of presentation could be reflected in the section's title.
→ Thank you for your suggestion. Section title changed to “Lipidomics and Metabolomics: Workflows and Analytical Techniques” .
In the same section, there is a mention of two types of metabolomics analysis: targeted and untargeted approaches. It would be beneficial to highlight that while the former involves hypothesis testing and finds application in clinical settings, the latter serves as a hypothesis generator and demands validation, making it a more labor-intensive process.
→ The following sentence is added to the manuscript.
“While the former is a hypothesis testing approach, the later is a hypothesis generating approach, involving discovery and validation of novel candidate biomarkers; however, the process is a labor-intensive.”
Between lines 210 and 215, the authors made a statement regarding the lack of established associations between low-abundant metabolites, such as DolPs and PIs, and diseases, but they did not provide any references to support this claim.
→ Thank you for your suggestion. Reference is provided now in the revised Manuscript
Furthermore, it remains unclear why these metabolites were mentioned. If the intent was to demonstrate how new techniques and technologies can enhance the sensitivity of methods used to quantify low-abundant metabolites, the phrase needs to be redone.
→ Thank you for your suggestion. The following sentence is added to the manuscript.
“Non-targeted assays are commonly used for high-throughput screening of small molecules in complex biological specimens. Despite its proven utility in the measurement of diverse molecules simultaneously, these assays are known to overlook low-abundance metabolites [66] (e.g., signaling lipids such as phosphatidyl inositol phosphates (PIPs), and eicosanoids and its metabolites). Analysis of such low-abundant metabolites needs targeted analysis, and may sometimes require additional selective sample enrichment/purification steps during sample preparation procedures prior to MS analysis. In recent years, several new MS-based analytical methods, and techniques ( e.g. [67, 68] ) have been developed. In the future, such analytical methods could be used to comprehensively characterize the homeostasis of these low-abundance analytes under health and disease conditions”
Table 1 is currently challenging to read and should be redesigned. For instance:
|
Disease |
References |
Analytical Platform |
Biomarkers |
Findings |
|
CVD |
WECAC Study [106] Lipid Study Karola Sudy |
LC-MS |
Ceramide and PC |
Ceramide-PC score as predictor of residual events risk in patients with CAD |
→ Thank you for your suggestion.
- Many clinical studies do not have abbreviations. Therefore, it will be difficult to write a table as mentioned above. Few studies with abbreviated names, all are mentioned as superscript abbreviations under Table 1.
- Table is modified and simplified for better understanding to the readers.
- One NMR based study was removed for clarity
- Abbrevivations are formatted for better alignment
Continuing with Table 1, diabetes is one of the listed conditions. Given that the title pertains to cardiovascular diseases, it would be better to eliminate these mentions from the table.
→ Thank you for your suggestion. The studies based on diabetes are removed.
There is no connection between the content found on page 8, lines 308-314, and the content located on page 11, lines 326-339.
→Thank you for your suggestion. The paragraph is moved to in connection with content now.
Reviewer 2 Report
Comments and Suggestions for Authors
The paper “Blood-derived lipid and metabolite biomarkers in cardiovascular research from clinical studies: a recent update” summarizes promising CVD biomarkers in human blood plasma /serum samples.
In the result section, the authors write a lot about lipidomic and metabolomic techniques and little attention is paid to clinical studies on the association of plasma trimethylamine N-oxide levels, elevated levels of acylcarnitines, or tryptophan degradation as a potential biomarkers for CVDs. This could be corrected.
Author Response
It is an insightful comment to improve the review-discussion. Thank you. Accordingly, we have added the table references which point out the association of these various metabolites in the conclusions section,. We also mentioned that these metabolites should be observed as a whole panel in clinical studies to uncover molecular mechanisms causing CVD.
Round 2
Reviewer 1 Report
Comments and Suggestions for Authors
The authors conducted a good review, making the text more fluid and enhancing the understanding of the subject.